# Resveratrol and (-)-Epigallocatechin-3-gallate Regulate Lipid Metabolism by Activating the AMPK Pathway in Hepatocytes

**DOI:** 10.3390/biology13060368

**Published:** 2024-05-23

**Authors:** Huanbin Wang, Yu An, Shahid Ali Rajput, Desheng Qi

**Affiliations:** 1Department of Animal Nutrition and Feed Science, College of Animal Science and Technology, Huazhong Agricultural University, Wuhan 430070, China; whbin@webmail.hzau.edu.cn (H.W.); ay2000@webmail.hzau.edu.cn (Y.A.); 2Department of Animal and Dairy Sciences, Faculty of Veterinary and Animal Sciences, Muhammad Nawaz Shareef University of Agriculture, Multan 60000, Pakistan; shahid.ali@mnsuam.edu.pk

**Keywords:** resveratrol, (-)-Epigallocatechin-3-gallate, LMH cells, AMPK, lipid metabolism, antioxidant capacity, cholesterol, fatty acid

## Abstract

**Simple Summary:**

Excessive synthesis of liver lipids can lead to various diseases, such as fatty liver disease and hyperlipidaemia. This study investigated the effects of resveratrol (Res) and (-)-Epigallocatechin-3-gallate (EGCG) on lipid synthesis in liver cells. The results showed that both resveratrol and EGCG can inhibit cholesterol synthesis in liver cells by activating Adenosine 5′-monophosphate (AMP)-activated protein kinase (AMPK). Therefore, this study can provide a reference for the study of lipid metabolism.

**Abstract:**

The purpose of this study was to explore the effects of Res and EGCG on cell growth, cellular antioxidant levels, and cellular lipid metabolism in hepatocytes. In this experiment, leghorn male hepatoma (LMH) cells were used as hepatocytes. The results showed that 6.25–25 μM Res and EGCG had no adverse effects on cell viability and growth. Meanwhile, with the increasing dosage of Res and EGCG, the contents of total cholesterol (TC), total glyceride (TG), and malondialdehyde (MDA) in hepatocytes decreased significantly (*p* < 0.05), while the contents of glutathione peroxidase (GSH-Px), total superoxide dismutase (T-SOD), and catalase (CAT) increased significantly (*p* < 0.05). In addition, western blot results showed that Res and EGCG could significantly increase the expression of p-AMPK protein and reduce the expression of 3-hydroxy-3-methylglutaryl-CoA reductase (HMGCR) protein in hepatocytes (*p* < 0.05). Moreover, q-PCR results showed that with the increase in Res and EGCG, the expression of cholesterol- and fatty acid synthesis-related genes decreased significantly (*p* < 0.05). In conclusion, Res and EGCG can increase the antioxidant capacity of hepatocytes and reduce the synthesis of TC and TG in hepatocytes by activating AMPK, thereby regulating lipid metabolism in hepatocytes.

## 1. Introduction

The liver is a vital organ responsible for lipid synthesis in humans and animals [1]. The liver is composed of hepatocytes, and abnormal lipid metabolism in hepatocytes can lead to various diseases [1,2]. Fat and cholesterol are important components of lipids [2]. When people and animals suffer from excess nutrition or stress, the excess energy in the body will be converted into fat and deposited in the liver, resulting in the occurrence of fatty liver syndrome, which will adversely affect the health of people and animals [3,4]. In addition, the liver is the main organ responsible for cholesterol synthesis [5]. After synthesis, cholesterol will be transported to the surrounding tissues to play various roles [6]. Increased cholesterol synthesis usually leads to the occurrence of hyperlipidaemia [6]. Therefore, reducing cholesterol synthesis in hepatocytes can prevent human hyperlipidaemia-related diseases [7]. It can be seen that regulating the anabolism of fatty acids and cholesterol in hepatocytes is of great significance for human health; therefore, this study was conducted to screen for exogenous additives that can regulate liver cell lipid synthesis.

AMP-activated protein kinase (AMPK) is a serine/threonine protein kinase, a switch involved in regulating cellular energy metabolism [8]. When exogenous substances activate AMPK, catabolic life activities in cells are enhanced to increase the ATP supply, and anabolic life activities are inhibited to reduce energy consumption [8]. Numerous studies have shown that when AMPK is activated, the synthesis of cholesterol and TG in cells is inhibited [8,9,10]. Hwang et al. (2013) reported that S-Allyl cysteine (SAC) can activate AMPK to reduce triglyceride (TG) accumulation in HepG2 cells [9]. Wang et al. (2023) reported that neferine (NEF) could reduce the accumulation of total cholesterol (TC) and TG in the livers of non-alcoholic fatty liver disease model mice by activating AMPK [10]. At present, there are few studies on the AMPK pathway in LMH cells and a study of AMPK and lipid metabolism in LMH cells has not been reported.

Res is a non-flavonoid polyphenol compound with the efficacy of anti-oxidative stress and cardiovascular protection [11]. There are numerous reports that Res can reduce the synthesis of TG and TC in many cells [12,13]. In addition, Price et al. (2012) reported that Res could activate AMPK, thereby improving mitochondrial function in mice [12]. Wei et al. (2021) reported that Res could activate AMPK, thereby improving lipid accumulation in human SZ95 sebocytes [13].

EGCG is the main component of tea polyphenols and belongs to the class of catechins, which have anti-oxidative stress and anti-atherosclerotic effects [14]. A study reported that EGCG could activate AMPK in the subcutaneous and epididymal adipose tissue of mice, thereby reducing cholesterol in serum and the accumulation of lipids in tissues around the body [15]. Another study revealed that EGCG can activate AMPK in human hepatoma cells, inhibiting the synthesis of proteins and lipids [16].

LMH cells are derived from chicken liver. Due to the demand for egg production in chickens, their lipid metabolism is relatively vigorous, so LMH cells are often used as research subjects to study lipid metabolism [17]. At present, the effect of Res and EGCG on lipid metabolism in LMH cells has not been reported. Numerous studies have shown that both Res and EGCG can activate the phosphorylation of AMPK. When AMPK phosphorylation is activated, intracellular TG and TC synthesis is inhibited. It is currently unknown whether Res and EGCG can regulate lipid metabolism in LMH cells by activating AMPK. This study found that both Res and EGCG could inhibit lipid synthesis in LMH cells by activating AMPK, filling the gap in this area of research. Therefore, this study can provide a reference for related studies on lipid metabolism.

## 2. Materials and Methods

### 2.1. Materials and Reagents

Dmem/f12 medium was purchased from Solarbio Biotechnology Co., Ltd. (Beijing, China). Super-grade foetal bovine serum was purchased from Beyotime Biotechnology Co., Ltd. (Wuhan, China). Res (cas:501-36-0, molecular weight 228.24, purity 99%) and EGCG (cas:989-51-5, molecular weight 458.37, purity 99%) were purchased from Shanghai Macklin Biochemical Technology Co., Ltd. (Shanghai, China). Both Res and EGCG were dissolved in DMSO.

### 2.2. Cell Culture

LMH cells were purchased from Huzhen Industrial Co., Ltd. (Shanghai, China). Before the experimental treatment, LMH cells were cultured in dmem/f12 medium supplemented with 1% penicillin/streptomycin and 10% extra-grade foetal bovine serum in 0.1% gelatine-coated culture flasks [18]. The culture flask was placed in a CO_2_ cell incubator (Boxun Medical Biological Instrument Co., Ltd., Shanghai, China) in an atmosphere of 95% air and 5% CO_2_ at 37 °C with 70–80% humidity [18].

### 2.3. Cell Viability Assay (CCK-8)

Cell viability was determined using a CCK-8 Kit (Biosharp Biotechnology Co., Ltd., Wuhan, China). LMH cells were transferred to 96-well plates for culture. After normal culture of LMH cells for 12 h, Res and EGCG were added to the culture medium to final concentrations of 6.25, 12.5, 25, 50, and 100 μM. The cells were incubated for another 24 h and 48 h. Then, 10 μL CCK-8 solution was added to each well. After a further 1 h of incubation in the cell incubator, the absorbance value of each well was determined at 450 nm. After the absorbance value of the blank group is subtracted from those of the treatment and control groups, the ratio of absorbance values of the treatment and control groups is the cell viability. The test was performed as outlined in the manufacturer’s instructions.

### 2.4. Cell Growth Trends

Based on the results of cell viability testing, doses of 6.25, 12.5, and 25 μM were selected for subsequent studies. The follow-up study was divided into eight groups, namely the control group, three Res treatment groups (medium supplemented with 6.25, 12.5, and 25 μM Res), three EGCG-treated groups (medium supplemented with 6.25, 12.5, and 25 μM EGCG), and the composite group (medium supplemented with 12.5 μM Res and 12.5 μM EGCG). The cell state was observed and photographed with an inverted microscope (Olympus, Tokyo, Japan) at 24 h and 48 h.

### 2.5. Determination of Cellular TC and TG Content

In the beginning, there were about 3 × 10^6^ cells in the culture bottle. After the cells of the eight groups were cultured for 12 h, the corresponding concentrations of Res and EGCG were added to the culture medium, and the culture was continued for 48 h. Then, the cells in the culture bottle were used to determine TC and TG. The TC and TG contents of LMH cells were determined using commercial kits from Nanjing Jiancheng Biotechnology Co., Ltd. (Nanjing, China) according to the manufacturer’s instructions. Cells were obtained using the same method to detect the antioxidant, gene, and protein expression.

### 2.6. Determination of Cellular Antioxidant Levels

GSH-Px, MDA, CAT, and T-SOD were determined using the corresponding commercial kits from Nanjing Jiancheng Biotechnology Co., Ltd. (Nanjing, China) according to the manufacturer’s instructions.

### 2.7. RNA Extraction and cDNA Synthesis

Trizol reagent (Thermo Fisher Scientific Co., Ltd., Shanghai, China) was used to extract total RNA from LMH cells according to the manufacturer’s instructions. Abscript III RT master mix for qPCR with gDNA remover reagent (ABclonal Biotechnology Co., Ltd., Wuhan, China) was used for reverse transcription to synthesise cDNA according to the manufacturer’s instructions.

### 2.8. Real-Time q-PCR

Real-time q-PCR was performed to detect gene expression using 2X Universal SYBR Green Fast qPCR Mix reagent (ABclonal Biotechnology Co., Ltd., Wuhan, China) and the ABI QuantStudio^TM^ 6 Flex PCR instrument (Applied Biosystems, Foster City, CA, USA), according to the manufacturer’s instructions. The gene names and primer sequences are shown in Table 1. The β-actin gene was used as an internal control, and relative gene expression was analysed by the 2^−ΔΔCt^ method.

### 2.9. Western Blotting Analysis

The cells were lysed with high-efficiency RIPA cell lysate (Solarbio Biotechnology Co., Ltd., Beijing, China). The lysate was precooled before use, and protease and phosphatase inhibitors were added to the lysate and mixed evenly. After the cells were fully lysed, the mixture was placed at 4 °C at 12,000× *g* for 15 min. The supernatant was collected, the protein was denatured with a protein loading buffer, and the protein concentration of the denatured sample was 2 μg/μL. An equal amount of protein in the sample was separated by 8–10% SDS-PAGE, and then the protein in the gel was transferred to a polyvinylidene fluoride (PVDF) membrane. After blocking with 5% nonfat milk powder or bovine serum albumin (BSA) for 3 h at room temperature, the primary antibody (ABclonal Biotechnology Co., Ltd., Wuhan, China) was incubated overnight at 4 °C, and then the corresponding secondary antibody (ABclonal Biotechnology Co., Ltd., Wuhan, China) was incubated for 1 h at room temperature. Western blots on PVDF membranes were detected with the beyoECL Plus Kit (Beyotime Biotechnology Co., Ltd., Wuhan, China). The results were quantified by the grey value of the western blot, and the GAPDH protein was used as an internal control. The results of the protein blot experiment was shown in Appendix A.

### 2.10. Statistic Analysis

SPSS 25.0 (IBM, Armonk, NY, USA) was used to process the data, and the results were expressed as mean ± SEM. One-way ANOVA was used to compare the results, and a *t*-test was used to evaluate the difference between the two groups of data. ^a,b,c,d,e,f^ Different lowercase letters indicate significant differences between the groups (*p* < 0.05).

## 3. Results

### 3.1. Effects of Res and EGCG on Cell Viability and Cell Growth

When Res and EGCG were added at 6.25–25 μM, the two compounds had no adverse effect on the cell viability of LMH cells for 24 h (Figure 1A). When the addition of Res and EGCG was greater than 25 μM, the cell viability of LMH cells decreased significantly at 24 h. The trend for the change in cell viability at 48 h was close to that at 24 h (Figure 1A). Three different doses of Res and EGCG did not adversely affect the growth status of cells (Figure 1B,C).

### 3.2. TC and TG Content

With the gradual increase in Res and EGCG, the content of TC and TG in LMH cells gradually decreased (Figure 2). The larger the dose, the greater the decrease in TG and TC content, which indicates the existence of a dose-dependent effect. And the difference was significant when the addition of Res and EGCG reached 12.5 μM and 25.0 μM (*p* < 0.05) (Figure 2). The TC and TG contents in the cells of the compound group were lower than those in the control group but higher than those of the Res or EGCG groups with 25 μM added alone (*p* < 0.05) (Figure 2).

### 3.3. Changes in Antioxidant Capacity

With the gradual increase in Res and EGCG, the contents of GSH-Px, CAT, and T-SOD in LMH cells slightly increased, while the content of MDA gradually decreased (Figure 3). When the addition of the two compounds reached 12.5 μM and 25.0 μM, the contents of GSH-Px, MDA, and CAT changed significantly (*p* < 0.05), and when the addition amount reached 25 μM, the change in T-SOD was significantly different (*p* < 0.05) (Figure 3).

### 3.4. Changes in Gene and Protein Expression Levels of the AMPK/HMGCR Pathway

Res and EGCG had no significant effect on *AMPK* gene mRNA and protein expression in LMH cells (Figure 4A). The expression of p-AMPK protein gradually increased with the increase in Res and EGCG, and when the addition level reached 12.5 μM and 25.0 μM, the difference was significant (*p* < 0.05) (Figure 4B). At the same time, the mRNA and protein expression of the *HMGCR* gene decreased with the increase in Res and EGCG addition, and when the addition level reached 25 μM, the differences were significant (*p* < 0.05).

### 3.5. Expression Changes of Cholesterol Synthesis-Related Genes

When the addition of Res and EGCG was 12.5 μM and 25.0 μM, the mRNA expression levels of *SREBP2*, *FDPS*, and *LSS* were significantly down-regulated (*p* < 0.05) (Figure 5). When the addition of Res and EGCG was at 25 μM, the mRNA expression of *MVK* decreased significantly (*p* < 0.05) (Figure 5). When the addition of Res was at 25 μM, the expression of *FDFT1* decreased significantly (*p* < 0.05) (Figure 5). With the increase in the dosage, the expression of *SQLE* decreased, but the difference was not significant (*p* > 0.05) (Figure 5).

### 3.6. Expression Changes of Fatty Acid Synthesis-Related Genes

With the gradual increase in Res and EGCG, the mRNA expression of fatty acid synthesis-related genes showed a downward trend (Figure 6). When the addition of Res and EGCG reached 25 μM, the mRNA expression of the *ACC*, *FASN*, and *SREBP1* genes was significantly different (*p* < 0.05) (Figure 6).

## 4. Discussion

Cell viability and cell growth status are important indicators that reflect the health of cells [19]. Currently, there is no report on the effect of Res and EGCG on the viability of LMH cells. Chang et al. (2013) reported that 10 μM Res or below could increase the cell viability of human PC-3 cells [20]; meanwhile, Yang et al. (2019) reported that 10 μM Res or above will adversely affect the cell viability of melanoma cells [21]. Yu et al. (2022) reported that 10 μM and below EGCG had no adverse effect on cell viability and cell cycle of MARC-145 and PAM cells [22], and Lin et al. (2012) reported that 24 μM EGCG or below had no adverse effect on the cell viability of mouse 3T3-L1 cells [23]. Meanwhile, Yu et al. (2007) reported that 10 μM and above EGCG adversely affect the cell viability of PC-3 cells [24]. According to previous reports, different types of cells respond differently to Res and EGCG. The present study found that when the concentration of Res and EGCG was not more than 25 μM, the cell viability of LMH cells increased slightly, and when the addition amount of both was more than 25 μM, the cell viability decreased significantly, so it was presumed that there was a Hormesis dose effect on the effects of Res and EGCG on LMH cells [25]. The trend of changes in cell viability at 48 h was similar to that at 24 h, this may be due to a decrease in the time-dependent response of LMH cells to the two additives. In order to perform a follow-up study of Res and EGCG on lipid metabolism without affecting the growth of LMH cells, the dose levels 6.25, 12.5, and 25 Μm of the two additives were selected. The results for the cell growth state also proved that when the addition of Res and EGCG did not exceed 25 μM, there was no adverse effect on the growth of LMH cells.

LMH cells originate from the liver, an essential organ for lipid synthesis, and the synthesis of TC and TG is an important indicator to reflect the lipid synthesis status of LMH cells [1,2,8,9,10]. Abnormal synthesis of TG and TC in liver cells can lead to various diseases, such as fatty liver syndrome and hyperlipidaemia [26]. The effects of Res and EGCG on the synthesis of TC and TG by LMH cells have not been reported. Previous studies have shown that adding Res to the diet of laying hens can significantly reduce the content of TC and TG in their serum [27], and adding 240 mg/kg Res to the diet of rats with gestational diabetes can significantly reduce the content of TC and TG in their serum [28]. At the same time, Gan et al. (2021) reported that EGCG could reduce the content of TG and TC in serum and reduce the formation of lipid droplets in the livers of mice with alcoholic fatty liver [29]. Li and Wu (2018) reported that adding 50 mg/kg EGCG to the diet of hyperlipidaemic rats could significantly reduce the content of TC and TG in serum and alleviate liver injury [30]. Most of the TC and TG in serum originate from the liver [1]. It can be seen that Res and EGCG have the efficacy of reducing the synthesis of TC and TG in the liver. The results of this study found that when 12.5 μM and 25.0 μM Res and EGCG were added, the intracellular TC and TG contents of LMH cells also decreased significantly, indicating that Res and EGCG can inhibit lipid synthesis in LMH cells.

Oxidative stress is closely related to lipid metabolism in hepatocytes and is considered a major factor leading to non-alcoholic fatty liver injury and disease progression [31,32]. Previous studies have shown that oxidative stress can promote the generation of fat in mouse adipocytes, inhibit fat decomposition, and thus contribute to lipid accumulation [31]. At present, the effects of Res and EGCG on the antioxidant capacity of LMH have not been reported. GSH-Px is a widely distributed peroxidase. GSH-Px in hepatocytes can catalyse GSH to participate in peroxidation reactions, scavenging peroxides and hydroxyl radicals to reduce lipid peroxidation [33]. MDA is the final product of lipid peroxidation, which can reflect the degree of lipid peroxidation in cells [33]. T-SOD can scavenge superoxide anion free radicals, thus protecting cells from damage [34]. CAT can quickly remove the residual hydrogen peroxide in the body to protect cells from the toxicity of H_2_O_2_ [34]. The content changes of these four substances can rapidly reflect the antioxidant level of cells or the body [33,34].

Previous studies have shown that Res and EGCG have antioxidant effects in vivo and in vitro [27,35,36,37]. Feng et al. (2017) reported that the addition of Res to the diet could significantly increase the content of GSH-Px in serum and reduce the content of MDA in the serum of laying hens [27]. Wu et al. (2022) reported that Res could increase the contents of GSH-Px, CAT, and SOD in the ovaries of aged rats, reduce the content of MDA, and thus reduce oxidative stress damage to the ovaries [35]. The effect of EGCG on oxidative damage has also been studied [36,37]. Karamese et al. (2016) reported that 400 μM EGCG could significantly increase the contents of GSH-Px and CAT in human liver cancer cells (Hep3B), thereby alleviating lipopolysaccharide-induced hepatotoxicity [36]. Li et al. (2019) reported that after intravenous injection of EGCG in mice treated with free fatty acids (FFAs), the content of MDA in the serum of model mice decreased significantly, and the contents of SOD and GSH-Px increased significantly [37]. In this study, we found that Res and EGCG could increase the content of GSH-Px, T-SOD, and CAT in LMH cells, reduce the content of MDA, and improve the antioxidant capacity of LMH cells, which was consistent with the previous results.

AMPK is the molecular switch of energy regulation in the organism, and the organism’s life processes, such as glucose metabolism, protein synthesis, and lipid metabolism, are regulated by AMPK [8,9,10]. When AMPK is activated, ATP-consuming life activities, such as lipid and protein synthesis, will be inhibited [8,9,10]. HMGCR is one of the rate-limiting enzymes in the process of cholesterol synthesis, regulating the biological process of the mevalonate pathway, and cholesterol belongs to the category of lipids, so HMGCR is also regulated by AMPK [6,8]. Previous studies have shown that the loss of AMPK in regulatory T cells (Tregs) can promote the expression of HMGCR [38], while the expression of HMGCR and hmgcs genes in HepG2 cells treated with AICAR (AMPK activator) decreased significantly [39]. At present, the effects of Res and EGCG on the AMPK/HMGCR pathway in LMH cells have not been reported. Wei et al. (2021) reported that Res can inhibit the inflammation of human SZ95 sebaceous cells by activating AMPK [13], and Timmers et al. (2011) reported that Res can activate AMPK in human muscle and reduce the content of lipids and inflammatory markers in the liver [40]. Meanwhile, Yang et al. (2016) reported that EGCG can activate AMPK in the liver, muscle, and white adipose tissue and participate in the metabolic regulation of fatty acids [41]. Moreover, Li et al. (2018) reported that EGCG could reduce the degree of obesity in mice by activating AMPK, inhibiting the expression of fatty acid synthesis-related genes, and promoting the expression of lipolysis-related genes in mice on a high-fat diet [15]. The results of this study showed that after Res and EGCG treatment of LMH cells, AMPK was activated, and the mRNA and protein expression of HMGCR were significantly decreased, indicating that Res and EGCG can regulate the AMPK/HMGCR pathway in LMH cells, which is similar to the results of previous studies. After the activation of AMPK in LMH cells, the lipid synthesis of LMH cells was inhibited, which may be the main reason for decreasing TC and TG content in LMH cells.

Sterol regulatory element binding protein 2 (*SREBP2*) is another key regulator of cholesterol synthesis in the body, which occurs in the endoplasmic reticulum, followed by processing and activation in the Golgi [6]. Baselga-Escudero et al. (2014) reported that after HepG50 cells were treated with 2 μM Res and EGCG for 1 h, the expression of the *SREBP2* gene in HepG50 cells had a downward trend, but it was not significant [42]. In this study, we found that when LMH cells were treated with 6.25 μM Res or 12.5 μM EGCG, the expression of the *SREBP2* gene also showed a downward trend, and it was also not significant, which was the same as the research results. When the addition of Res and EGCG continued to rise, the expression of the *SREBP2* gene decreased significantly, indicating that the expression of the *SREBP2* gene may also be regulated by AMPK, thereby affecting the synthesis of TC in LMH cells, which is consistent with the results of Tang et al. (2016) [43].

*FDPS*, *FDFT1*, *MVK*, and *LSS* are important genes in the process of cholesterol synthesis [7]. Miao et al. (2016) reported that DMU-212, a derivative of Res, could regulate the expression of the *MVK* gene in vascular endothelial cells [44]. Li and Wu (2018) reported that EGCG could inhibit the increased expression of *FDPS* in hyperlipidaemic rats [30]. In addition, Ge et al. (2014) reported that EGCG could inhibit the enzymatic activity of *MVK* in vitro [45]. The effects of Res and EGCG on the expression of cholesterol synthesis-related genes in LMH cells have not been reported. Previous studies have shown that when AMPK is activated, the expression of genes related to cholesterol synthesis, such as *FDPS*, will be inhibited [8,9,10]. The present study showed that Res and EGCG could inhibit the expression of genes related to cholesterol synthesis in LMH cells, which may be associated with the activation of the AMPK/HMGCR pathway by Res and EGCG. When AMPK is activated, cholesterol synthesis is inhibited, and the mRNA expression of genes related to cholesterol synthesis in LMH cells is down-regulated by positive feedback regulation.

*ACC*, *SCD1*, *FASN*, and *SREBP1* are essential genes for hepatocyte fatty acid synthesis [46]. The effects of Res and EGCG on the expression of fatty acid synthesis-related genes in LMH cells have not been reported. Sun et al. (2019) reported that Res could reduce the mRNA expression levels of *ACC* and *FAS* in human visceral preadipocytes (HPA-v) [47], while Gracia et al. (2016) reported that Res could reduce the expression of *SREBP1* protein and *FASN* gene in obese mice [48]. Meanwhile, Li et al. (2018) reported that EGCG could significantly reduce the expression of fatty acid synthesis-related genes *ACC1*, *SCD1*, *FAS*, and *SREBP1* in the epididymal adipose tissue of mice [15]. In this study, our results showed that Res and EGCG could significantly down-regulate the mRNA expression of *ACC*, *FASN*, and *SREBP1* genes in LMH cells, and the results were similar to those of previous studies. The synthesis of fatty acids is also regulated by AMPK [8]. Res and EGCG can activate AMPK in LMH cells, resulting in the inhibition of fatty acid synthesis. Also affected by positive feedback regulation, the expression of fatty acid synthesis-related genes decreased.

From the results of this experiment, the simultaneous addition of 12.5 μM Res and EGCG did not decrease the content of TC and TG in LMH cells by more than the addition of 25 μM Res or EGCG alone. This result was also obtained for mRNA and protein expression of lipid metabolism-related genes. This indicates that there is no synergy between Res and EGCG.

The effects of Res and EGCG on lipid metabolism in LMH cells are shown in Figure 7. After treatment with Res or EGCG, the antioxidant level of LMH cells increased, while AMPK was activated, and the expression of phosphorylated AMPK protein increased. The cholesterol and fatty acid synthesis genes were inhibited, eventually leading to a decrease in cholesterol and triglyceride synthesis in LMH cells. However, this study did not delve deeper into the upstream pathways of Res and EGCG activation of AMPK phosphorylation, nor did it delve deeper into the signalling pathways involved in oxidative stress, which is a limitation of this study. In the future, in-depth research can be conducted in these two areas.

## 5. Conclusions

Presumably, this study was the first to explore the effects of Res and EGCG on lipid metabolism in LMH cells and mainly found that Res and EGCG could increase the antioxidant capacity of LMH cells and reduce the synthesis of lipids in LMH cells. Its mechanism of action is that Res and EGCG can activate AMPK to inhibit the expression of lipid synthesis-related genes and proteins in LMH cells. This study can provide a reference for studying liver cell lipid metabolism and related disorders.

## Figures and Tables

**Figure 1 biology-13-00368-f001:**
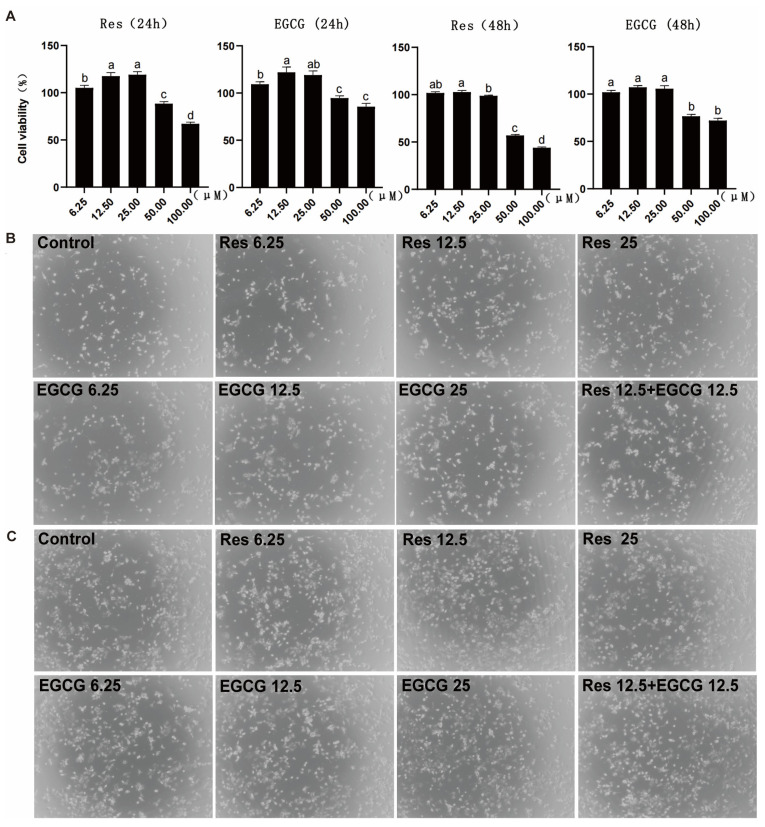
The effects of Res and EGCG on the viability and growth status of LMH cells. (**A**) Changes in cell viability at 24 and 48 h (n = 10). ^a,b,c,d^ Different lowercase letters indicate significant differences between the groups (*p* < 0.05). (**B**) The 24 h growth status of LMH cells. (**C**) The 48 h growth status of LMH cells.

**Figure 2 biology-13-00368-f002:**
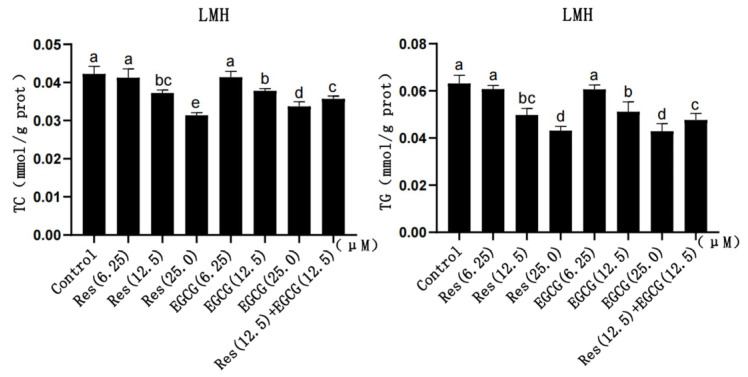
The effects of Res and EGCG on TC and TG levels in LMH cells (n = 6). ^a,b,c,d,e^ Different lowercase letters indicate significant differences between the groups (*p* < 0.05).

**Figure 3 biology-13-00368-f003:**
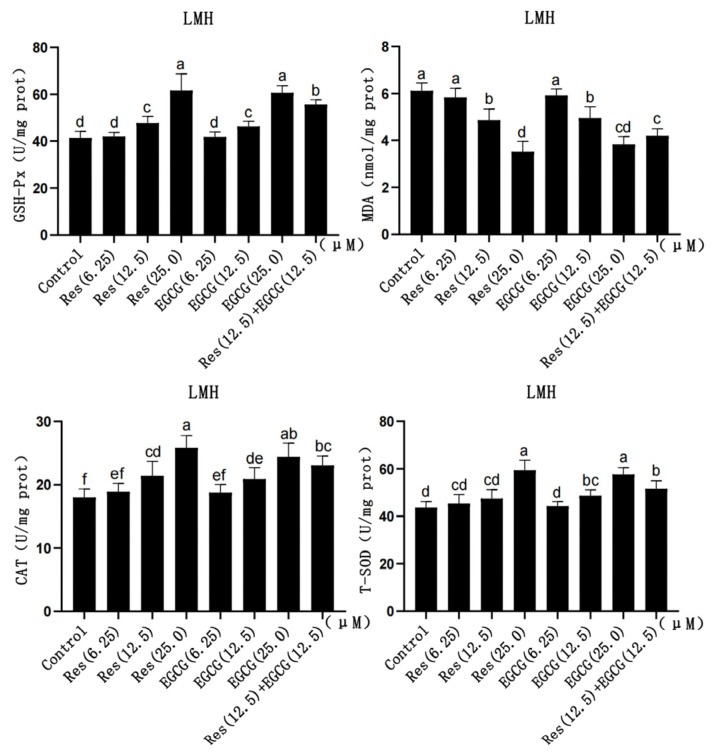
The effects of Res and EGCG on the antioxidant level of LMH cells (n = 6). ^a,b,c,d,e,f^ Different lowercase letters indicate significant differences between the groups (*p* < 0.05).

**Figure 4 biology-13-00368-f004:**
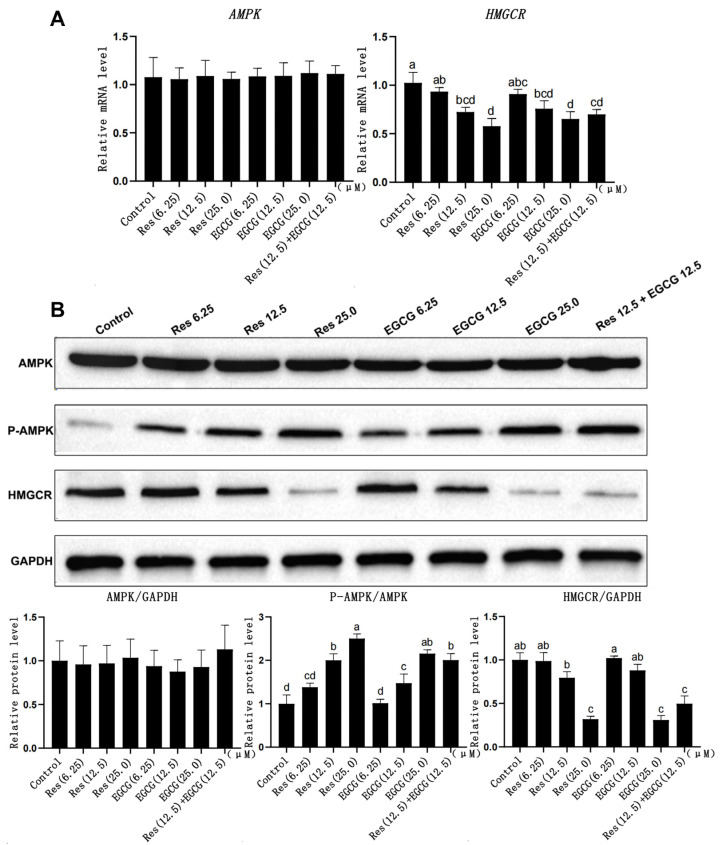
The effects of Res and EGCG on gene and protein expression levels in the AMPK/HMGCR pathway of LMH cells. ^a,b,c,d^ Different lowercase letters indicate significant differences between the groups (*p* < 0.05). (**A**) Changes in gene expression levels in the AMPK/HMGCR pathway (n = 6); (**B**) changes in protein expression levels in the AMPK/HMGCR pathway (n = 3).

**Figure 5 biology-13-00368-f005:**
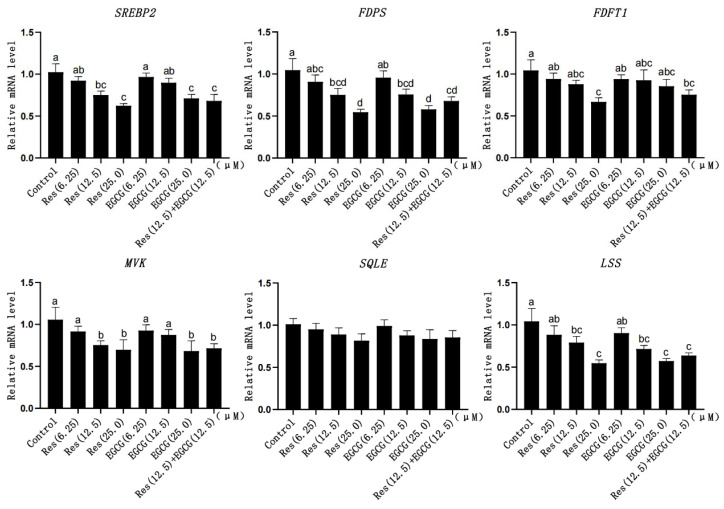
The effects of Res and EGCG on the expression of cholesterol synthesis genes in LMH cells (n = 6). ^a,b,c,d^ Different lowercase letters indicate significant differences between the groups (*p* < 0.05).

**Figure 6 biology-13-00368-f006:**
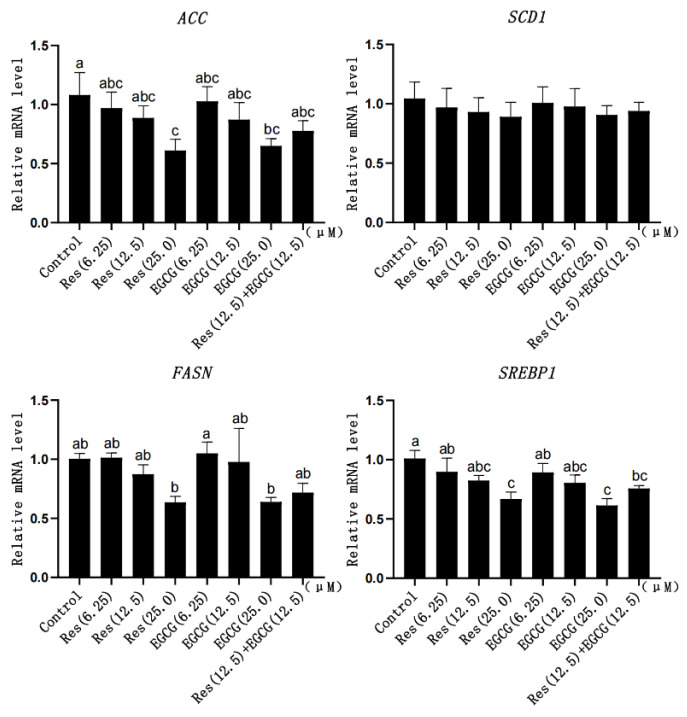
The effect of Res and EGCG on the expression level of fatty acid synthesis-related genes in LMH cells (n = 6). ^a,b,c^ Different lowercase letters indicate significant differences between the groups (*p* < 0.05).

**Figure 7 biology-13-00368-f007:**
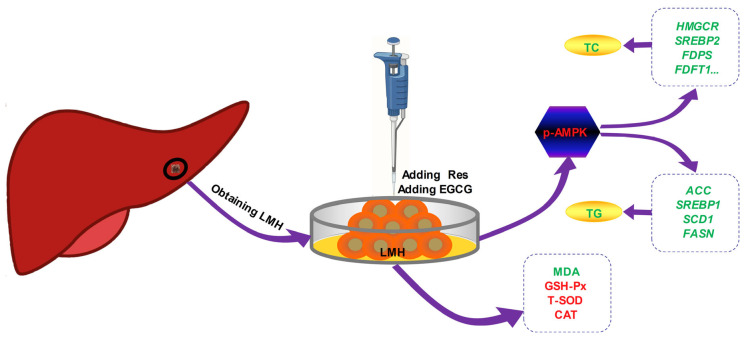
The effects of Res and EGCE on lipid metabolism in LMH cells. Red means up, green means down.

**Table 1 biology-13-00368-t001:** Primers for PCR ^1^.

Gene Name	Gene ID	Primer Sequence (5′-3′)	Product Size (bp)
*AMPK*	427185	F: CAAGCCCGCCAGATTCTTTT	184
		R: CGACTCTGACTCCGTATCCC	
*HMGCR*	395145	F: AGGAGCTTGCTGTGAAAACG	254
		R: AAGCTTTCACTTCTGCAGCC	
*SREBP2*	395304	F: AAGAGAAGATCCCACCCAGC	259
		R: TTCCTCTGGGTGCAGTACAG	
*FDPS*	425061	F: GGTTGGTGCATCGAGTTGTT	181
		R: GCCTGCAGTACTTCTTCAGC	
*FDFT1*	422038	F: GCCTTTCCCGACTCTTCTCT	152
		R: GGGCCAGAACTCTCTTCCTT	
*MVK*	768555	F: TCGGGTGTGGATAATGCTGT	159
		R: CTTCTCCTTAACCCCAGCCA	
*SQLE*	420335	F: TTGTGGGTTCAGGTGTCCTT	234
		R: CCTCCGACTTGCTCTCTAGG	
*LSS*	424037	F: CCAAGCAGATGACAGATGGC	280
		R: CAGATGGGGAAGACATTGCG	
*ACC*	396504	F: AAACTGATGGGGACGTGGAT	261
		R: ATGGAATGGCAGTGAGGTCA	
*SCD1*	395706	F: TTGCAAACTCCATGGCCTTC	193
		R: TCACTCAGGTCCAGCTTCTG	
*FASN*	396061	F: AGCTGGACTACATTGCCACT	238
		R: GTGTGCAGCAAAACAAACCC	
*SREBP1*	373915	F: GACCTCCAGCATCACCTCTT	290
		R: CCGACTTGTTGAGCTTAGCC	
*β-actin*	396526	F: AGTACCCCATTGAACACGGT	197
		R: ATACATGGCTGGGGTGTTGA	

^1^ Abbreviations: *AMPK*, adenosine 5′-monophosphate (AMP)-activated protein kinase; *HMGCR*, 3-hydroxy-3-methylglutaryl-CoA reductase; *SREBP2*, sterol regulatory element binding transcription factor 2; *FDPS*, farnesyl diphosphate synthase; *FDFT1*, farnesyl-diphosphate farnesyltransferase 1; *MVK*, mevalonate kinase; *SQLE*, squalene epoxidase; *LSS*, lanosterol synthase; *ACC*, acetyl-CoA carboxylase alpha; *SCD1*, stearoyl-Coenzyme A desaturase 1; *FASN*, fatty acid synthase; *SREBP1*, sterol regulatory element binding protein 1; *β-actin*, actin beta.

## Data Availability

Raw data are held by the author and may be available upon reasonable request.

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
