# Peer review of "Resveratrol and (-)-Epigallocatechin-3-gallate Regulate Lipid Metabolism by Activating the AMPK Pathway in Hepatocytes"

_biology, 2024, doi:10.3390/biology13060368_

Round 1

Reviewer 1 Report

Comments and Suggestions for Authors

This study examines the effects of Resveratrol (Res) and (-)- Epigallocatechin-3-gallate (EGCG) (mainly used as dietary supplements) on cell growth, cellular antioxidant levels, and cellular lipid metabolism in Leghorn Male Hepatoma (LMH) cells. The authors have studied the effects of RES and EGCG in terms of Total cholesterol (TC) and Triglyceride (TG) content in the LMH cells. They measured the levels of MDA (an indicator of oxidative damage to lipids) and looked at other markers of oxidative stress; GSH-Px, T- SOD, and CAT. They conclude that Res and EGCG can increase the antioxidant capacity of hepatocytes and reduce the synthesis of TC and TG in hepatocytes by activating AMPK.

Specific Comments:

Clarity and organization: The manuscript is written in a very simplistic and non-scientific manner. It is good to start with, but the authors should delve into the purpose of the study more deeply and with a more scientific approach. The manuscript would greatly benefit by dividing it into clearer subsections like:

For Introduction: The introduction is written in a very simplistic and non-scientific manner. It is good to start with, but the author should delve into the purpose of the study more scientifically. They abruptly mention the cholesterol content of egg yolk in the introduction without giving any specific context to their research. The introduction section should clearly describe the purpose of the study, why a model system (LMH cells) was selected,  and the parameters measured in the study. The introduction should also clearly mention the knowledge gap the undertaken study is filling.

For the Discussion section, clearer subsections are required: what you observe: effects of Res and EGCG on cell growth, TC, and TG. Please make a separate section on Oxidative stress: explain what biomarkers you choose to look at (give a very brief explanation) MDA, GSH, SOD, and CAT, then delve into what you found and relate to what is already known.

The AMPK pathway should be a separate subheading in the Introduction and Discussion. How do you think your observations relate to the AMPK pathway? Can you draw specific conclusions like what step of the pathway is affected?

Integration of results: The authors only report what previous studies have shown but do not interpret how what is already known affects their research or how their study contributes to the known scientific literature. Let the reader understand what knowledge gap your research fills is essential. It is equally important to point out the drawbacks/ biases in your research and give suggestions for future research.

Figures: For all the figures with histograms, please elaborate on the a,b,c, and d you use on top of them in the method section. Please show the stars of significance on them.

For Figure 1A: What was the value of control? It is not depicted in the graph. In line 108, you mention having a blank group and a control group; how are they different? What were their values?

What was the cell number you started with?

The authors conclude, ' This indicates no synergy between Res and EGCG, in line 385.' Does this mean that they affect different pathways?

Comments on the Quality of English Language

Minor corrections are required.

Reviewer 2 Report

Comments and Suggestions for Authors

General comment:

This paper needs an extensive revision of both the manuscript and the data, the style of writing is repetitive and devoid of detail, it is also not at all acceptable and deprives the work presented of the potential scientific significance it could have. I summarized my comments below but I believe that the scientific work presented would benefit from a complete rewrite.

General suggestions: the acronyms must be indicated at the first appearance

In materials and methods, it is not necessary to repeat the same initial sentence over and over again, but please modify the phrase with other ways of saying the same thing; it is never indicated the number of cells used and the purpose for which an assay or protocol is applied,  which should instead be the introductory phrase to these paragraphs and sub-paragraphs.

In the results: it is not necessary to start each paragraph with the same sentence in reference to the figure, insert in brackets the figure number. It is important that authors present the results with precision, details and indicating the purpose of each experimental set: the purpose that the data support or not. In addition, the words dose or below (or similar) are repeated everywhere in the text, but the range of doses used must always be specified with precision, and only in some cases, to avoid repetitions can be approximated in the manner aforementioned.

discussion: I would suggest the authors to revise it according to the suggestions above, in fact in this section the results should be discussed in the context of liver dysfunction introduced in the introduction. However, the context is no longer reflected in the results. Please avoid parts out of context and give a logical thread to the results.

review in detail:

line 50: relocate the sentence or contextualize, it would be more suitable to the materials and methods but if you want to insert here introduce it better.

line 56 TG acronym not presented the first time 

line 57: introduce and contextualize the list of evidences reported in support of experimental design, altrimetni remains a list disconnected from the body of the manuscript.

line 79/85: please rewrite, reduce and present in a more attractive way the experimental purpose, it is not necessary to introduce the techniques, this part must be placed appropriately in the section materials and methods.

paragraph 2.5-2.6 rewrite in a way more understandable to the reader, as mentioned above

157: specify the amount of proteins, please

Results: specifiy the range of tested doses and include in figure legends a clear description of the statistical analysis with the values of p value corresponding to each letter and the control on which the analysis was made  ( es: a:p<0.01 vs control; b...; c...)

Figure 1 A: Please chart the bar or a graphical reference of the control condition (untreated cells), against which to evaluate and compare the action of substances on proliferation, viability and growth.

line 177-178:The trend for the change in cell viability at 48 h was close to that at 24 h. the authors can describe this trend as a time-dependent decrease. Please discuss better.

Parag.3.1-3.2:these two paragraphs can be merged

Par 3.3: this paragrapher needs a better introduction, and please define in the text the doses and the identity of compounds group  to avoid confusion.

figure 2 : present these results better: Res A 25 um decrease compared to the control, and a gradual dose-dependent decrease is visible also with ECGG.Moreover, even if the compound group modulates this trend always remains lower than the level of control. This part should be highlighted in the manuscript and better presented.

Oxidative stress: Oxidative stress levels are evaluated directly and indirectly in the presence of exogenous stimuli. But there is no positive oxidative stress control, for example H2O2, that should be inserted.

 So the cells encroach and activate the signaling pathways involved in oxidative stress without an oxidative stimulus? Explain better in the manuscript. The work would benefit in being enriched with levels of oxidative stress not only basal but also the possible reduction of oxidative stress induced (eg treatment RES + H2O2, etc.).

Western blot quality must improved: the intensity of the blots total AMPK and GAPDH, overexposed, could affect the quantification submitted, since they do not allow to notice the differences between the different bands, if possible, please replace them.

Figure 7: this figure and the relative paragraph are not considered a result, but rather a scheme, a graphic abstract or a hypothetical mechanism, which supports the data presented, and then discussed. please,place this part in the introduction, discussion or as a graphical abstract.

Discussion: According with the general comment please revise this part, contextualize the data and discuss them avoiding evidence not related to the proposed study ( es: evidence on melanoma, cancer forms, etc.) I suggest to rewrite it in the best possible way.

Comments on the Quality of English Language

Please improve the quality of manuscript, enrich the vocabulary used avoiding to always use the same words. write in a clear but detailed English

Round 2

Reviewer 1 Report

Comments and Suggestions for Authors

This study examines the effects of Resveratrol (Res) and (-)- Epigallocatechin-3-gallate (EGCG) (mainly used as dietary supplements) on cell growth, cellular antioxidant levels, and cellular lipid metabolism in Leghorn Male Hepatoma (LMH) cells. The authors have studied the effects of RES and EGCG in terms of Total cholesterol (TC) and Triglyceride (TG) content in the LMH cells. They measured the levels of MDA (an indicator of oxidative damage to lipids) and looked at other markers of oxidative stress; GSH-Px, T- SOD, and CAT. They conclude that Res and EGCG can increase the antioxidant capacity of hepatocytes and reduce the synthesis of TC and TG in hepatocytes by activating AMPK.

The authors have made commendable efforts to accommodate all the requested changes. They have satisfactorily addressed all the queries. The Discussion section has significantly improved, which will likely lead to enhanced readership and stimulate thought-provoking research in the future.

This manuscript is acceptable after minor changes made to the overall English. For instance, in lines 117 and 118; begin a new sentence with a capital letter. It should be eight groups.

Comments on the Quality of English Language

Minor editing is required. For instance, in lines 117 and 118; begin a new sentence with a capital letter. It should be eight groups.

Reviewer 2 Report

Comments and Suggestions for Authors

I thank the authors for making the changes I suggested, so the paper has acquired more clarity and more scientific significance.

despite the paper has some limitations
